# Experimental Monitoring of Dynamic Parameters of the Sub-Ballast Layers as a Prerequisite for a High-Quality and Sustainable Railway Line

**Libor Ižvolt [1], Peter Dobeš [1], Zuzana Papánová [2,*] and Martin Mečár [1]**

1   Department of Railway Engineering and Track Management (DRETM), University of Žilina, Univerzitná 8215/1, 010 26 Žilina, Slovakia; libor.izvolt@uniza.sk (L.I.); peter.dobes@uniza.sk (P.D.); mecar@uniza.sk (M.M.)
2   Department of Structural Mechanics and Applied Mathematics (DSMAM), University of Žilina, Univerzitná 8215/1, 010 26 Žilina, Slovakia
*   Correspondence: zuzana.papanova@uniza.sk; Tel.: +421-415135607

**Abstract:** Monitoring dynamic load transfer from train traffic to sub-ballast layers is crucial for verifying the reliability and safety of railway lines, assessing the design cost-effectiveness and achieving minimum environmental impact. For this purpose, measurements in labs, in situ or modeling the influence of dynamic loads on the immediate and long-term roadway quality are often performed using suitable software. The available test sections enabled monitoring of the dynamic loads and optimizing the critical spots where increased dynamic effects from railway traffic may occur. The subject of this paper is the calibration of the sensors installed in the different test sections of the trans-European corridor number V. As a result, the necessary input parameters for the subsequent numerical modeling of the dynamic effects on the track substructure and vibration propagation on the available sections of the upgraded railway line were obtained. The sensor calibration was carried out on the experimental field, part of the Experimental Basis of the Department of Railway Engineering and Track Management. As part of the calibration, the sensitivity of the sensors embedded in the track bed to the applied dynamic loads resulting from the impact effects of the lightweight deflectometer was assessed. The result of the calibration was the demonstration of sufficient sensitivity of the sensors and their suitability for implementation in an actual railway track structure, with the aim of obtaining relevant values of the response of the sub-ballast layers to dynamic loads and assessing the operational impacts on the sustainable environment. Also, the main result of the research was the possibility of using the theoretical–experimental route to optimize the layers of the railway body.

**Keywords:** railway track; dynamic loading of the track; structural sub-ballast layers; diagnostics; lightweight deflectometer; pressure and acceleration sensors

## 1. Introduction

The railway vehicle roadway, encompassing rails, sleepers, track fittings, the track bed, sub-ballast layers and associated structures, needs to possess sufficient resilience against deformation and endure the impact of various adverse factors. Simultaneously, however, the permanent way is the source of many negative factors (especially noise and vibration), which must be eliminated as far as possible to respect sustainability principles. Throughout its operational lifespan, the railway line should have a minimal negative impact on the sustainable development and natural environment of the transversed area.

Rail transport faces competition from alternative modes of transport. The competitiveness of rail transport on national, regional and international levels must be seen in the context of sustainable development because competitiveness and sustainability are not mutually exclusive. To achieve sustainability, we must treat the environment in the context

of rail transport, even though it is the least environmentally damaging of all transport modes, as inseparable parts of a whole.

This must be done throughout the entire life cycle of a railway line, i.e., from the design phase of parts of the railway infrastructure (railway lines, railway stations, their facilities and equipment), including the necessary technical equipment (signaling and safety equipment, electrification), through the period of its construction and operation, to its reconstruction, modernization and eventual disposal. The beginning of the future railway infrastructure segment, the preparation of the project documentation, has a major impact on the sustainable development of the area of the project implementation and operation. The actual effects and impacts of the operation of the railway line or station on its surroundings must be identified at the design stage of the railway construction and its buildings or facilities to use environmentally tolerable practices, means, materials and technologies. In this context, it is significant to quantify and qualify the realistic stresses on the roadway for the specific type of rolling stock, the structural arrangement (track skeleton, sub-ballast layers) and the location of the railway line or station.

### 1.1. Research Background

The issue of stressing the trackbed layers has been investigated at the Department of Railway Engineering and Track Management (DRETM) in cooperation with other departments of the University of Žilina for a long time. The research activities initially focused on monitoring and subsequent numerical modeling of the effect of non-transport (climatic) loads on the construction of linear buildings [1–3]. Later, the research assessed the possibility of applying thermal insulation materials in the structural layers of linear buildings and, in some cases, also monitoring their deformation resistance (assessment of the static component of the traffic load) [4–7]. Subsequently, the research activities focused on transition zones (transition between the earthwork and the buildings of the sub-ballast layers) on the already upgraded parts of the railway infrastructure of the Slovak Railways [8,9].

Since 2021, the research of DRETM with the Department of Structural Mechanics and Applied Mathematics (DSMAM) has also focused on the monitoring of the traffic load, namely the influence of the dynamic effects of rolling stock and the propagation of the dynamic component of the traffic load in the transition zones of the railway lines and their adjacent sections. In doing so, it is necessary to observe both primary structural parts of the railway line, namely the railway superstructure and the substructure, as they form a single technical-physical unit. At the same time, both of these components of the railway line are equally important for developing a reliable and safe railway line that has a minimum negative impact on its surroundings.

For the implementation of numerical modeling of the propagation of the dynamic component of the traffic load in the trackbed, which has a major impact on the magnitude of noise emissions and vibrations spreading into the railway surroundings (and negatively affecting not only people and animals but also structures in its vicinity), pertinent input parameters are always needed. Figure 1 provides a location of three test sections prepared for this purpose. All the sections are located on the upgraded railway line, part of the trans-European corridor Va, and the dynamic load sensors are located in the transition zone and the subsequent part of the railway line running on the earthwork (embankment). Test section No. 1 (blue circle in Figure 1) is positioned behind the bridge structure in the vicinity of Railway Stop Milochov, test section No. 2 (green circle in Figure 1) in front of the bridge structure in the vicinity of Railway Stop Lučivná and test section No. 3 (purple circle in Figure 1) is located behind the bridge structure in front of Railway Station Svit. All localization characteristics are related to the track in the direction of its origin for standard railway operation (right-hand traffic).

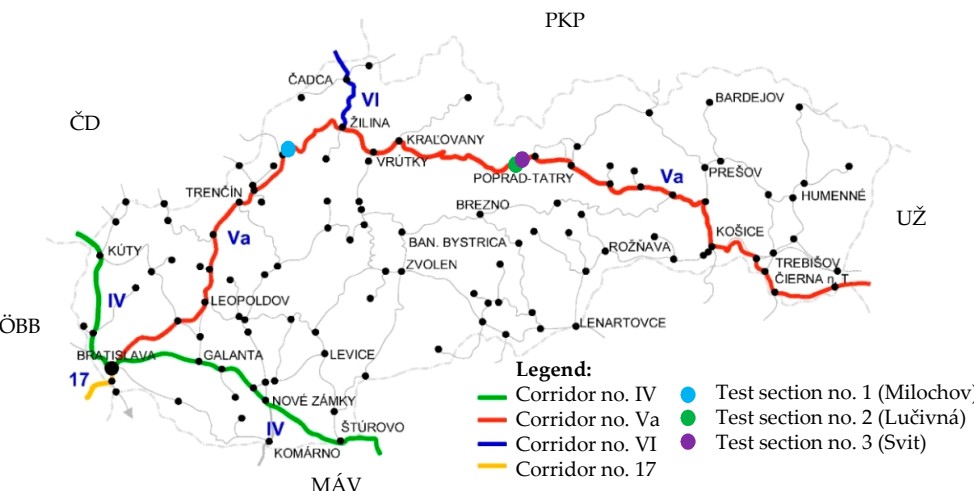

**Figure 1.** Location of test sections under construction.

Prior to establishing the designated test sections, it was imperative to assess the sensitivity and appropriateness of the measuring equipment, specifically pressure sensors and acceleration sensors. The sensors were initially planned and subsequently integrated into the relevant test sections. The characteristics of these sensors and the outcomes confirming their suitability constitute a focal point of discussion within this article.

The transition zones between the earthwork and the structures of the sub-ballast layers or artificial structures (tunnel, bridge, culvert, underpass) are critical places on the railway line due to the increased dynamic effects. If the gradual gradation of stiffness is not optimally resolved there, they become weak points in terms of the required quality of the track, as there is a gradual degradation of the designed track geometry. It results in increased dynamic effects of the rolling stock on the track skeleton, which causes increased noise and vibration emissions that spread to the railway surroundings, but also increased stresses on the structural layers of the trackbed and its weakest part, namely the subgrade surface, also with a negative impact on the track geometry quality.

### 1.2. State of the Art

Globally, several institutions are exploring the design and monitoring of the structural and material arrangement of transition zones between the earthwork and artificial structures of railway lines. The publication [10] presents new possibilities for monitoring the transition zones using the InSAR (Satellite Synthetic Aperture Radar) method. The results of settlement and settlement rate measurements in the transition zone using this method were also compared with the results obtained using the Digital Image Correlation (DIC) device and the measuring coach. The results of the three measurement techniques demonstrated a satisfactory correlation and indicated the possibility of using InSAR to monitor the quality of the structural condition of transition zones on a railway line. The condition of the structural elements of the superstructure and the trainsets running on the railway track considerably influences the magnitude of the dynamic effects transmitted from the train traffic to the structural layers of the transition zone. The design of a model for determining the dynamic effects arising from running trainsets with out-of-round wheels (different wheel curves) is presented in [11]. A numerical model demonstrating the assessment of the influence of the permanent way inhomogeneities on the magnitude of the emerging dynamic effects from running trainsets is presented in [12]. A summary of the problems arising within the transition zones, as well as the proposal of alternative solutions based on numerical modeling, is presented in [13], which provides a suitable basis for addressing the issue. The procedure for calculating the dynamic effects of the train passage and the response of the track to the load in question using a numerical model was presented in [14,15]. The specification of boundary conditions (material characteristics),

which have a significant influence on the dynamic response of the railway line, was carried out based on the developed numerical model characterized in [16].

Abroad, much attention is now dedicated to the sustainable development of rail transport and saving natural resources. A comprehensive analysis of the trends and challenges in railway sustainability, with a specific focus on materials and components of the permanent way and methods for sustainability assessment, is presented in [17]. The methods for reducing noise emissions and vibrations by rail traffic (construction and legal methods) and an innovative analysis for limiting their impact on natural habitats and forest animal populations are detailed in [18].

The analysis of the impact of technological innovation and modernization of railway infrastructure on ecological sustainability is presented in [19]. Methods of reducing the noise and vibration levels of railway transport to improve its environmental friendliness are presented in the publication [20]. In the study [21], an assessment of the performance of noise and vibration reduction methods (application of metamaterials, geosynthetics, subgrade improvement) in the vicinity of railway lines during their life cycle, which can increase railway industry sustainability, is carried out. This literature review indicates that the issue of permanent way quality and its impact on the generation of noise emissions and vibration is highly topical, and its solution requires diversified approaches to identify the magnitude of these negative impacts affecting sustainability. They rely not only on experimental measurements on operated railway lines but also on the use of computer technology, i.e., the possibility of modeling different cases of railway operation, railway line structures and their surroundings.

Thus, a prerequisite for obtaining relevant outputs from numerical modeling is to specify realistic input parameters and boundary conditions for the numerical model obtained (preferably) from experimental measurements. Therefore, also by the authors of this paper, several test sections (Figure 1) have been set up in which higher numbers of pressure and acceleration sensors (depending on the type of structure and the length of the transition region) were embedded at several locations and positions of their transition sections.

## 2. Materials and Methods

Before establishing the test sections, it was necessary not only to select suitable locations for monitoring the effects of railway traffic but also to select and calibrate suitable measuring and recording equipment. The calibration of the measuring equipment (in our case pressure and acceleration sensors) was performed within the experimental facility called the DRETM Experimental Stand [1]. The present experimental workstation has a steel frame with a movable crossbar, which can be used as a counterweight in the case of static load tests to determine the deformation resistance of various structural layers of linear transport structures. As the DRETM is currently engaged in research on the possibility of applying thermal insulation materials to the trackbed construction, in addition to the transition zones of the railway line, the design composition of the experimental field used for the verification of the suitability and calibration of the measurement equipment was supplemented with a foam glass aggregate, fr. 0/63 mm. This was done to obtain the necessary input parameters for the numerical modeling of the thermal regime of the trackbed construction with the thermal insulation material. The design and material composition of the experimental field, with the location of the measurement equipment and the implementation of the load tests (dynamic load tests), can be seen in Figure 2.

The individual structural layers of the experimental field were built on a subgrade surface of fine-grained soil (clay with gravel admixture). The deformation resistance (static modulus of deformation) of the subgrade surface, determined in a static load test performed according to [22], was in the range of $E_{def2} = 14$ to 15 MPa. The compaction, according to [23], was in the range of $E_{def2}/E_{def1} = 1.70$ to 1.90 (the maximum value allowed for fine-grained soils is 2.50).

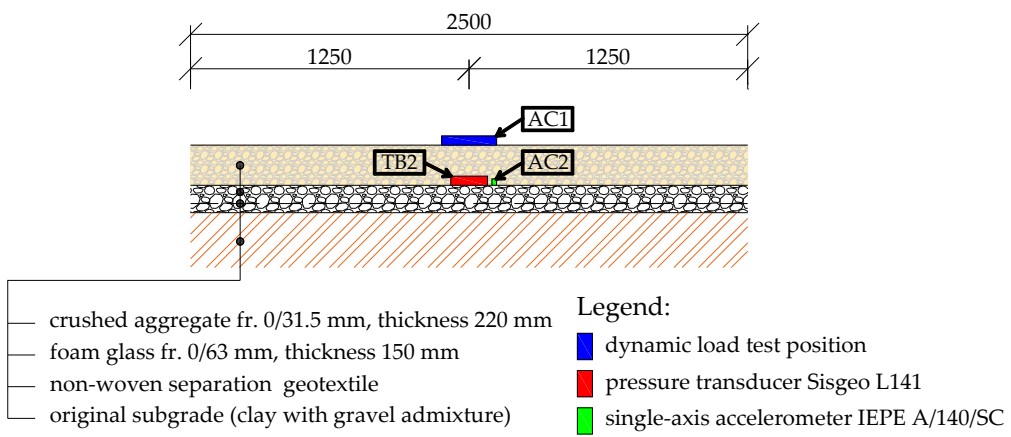

**Figure 2.** Experimental field.

The implementation of the experimental field and the measurements of the deformation resistance of the individual structural layers were carried out in March 2023. This period was selected because the lowest values of the deformation resistance are expected in the spring period, and for the dimensioning of the structural layers, it is necessary to be aware of the deformation resistance for the most adverse conditions. The materials used for the construction of the experimental field (crushed aggregate and foam glass) were stored in outdoor conditions during the winter period 2022/2023 to ensure their natural moisture content. The subgrade moisture content (determined near the experimental field using a device employing Time Domain Reflectometry (TDR)) was approximately 20% during the experimental measurements. The moisture content of the crushed aggregate and foam glass was determined after conducting the experimental measurements using a destructive method (by taking several samples and drying them). The moisture content of crushed aggregate (dry bulk density 1930 kg·m$^{-3}$) and foam glass (dry bulk density 135 kg·m$^{-3}$) was determined from several samples and was between 5 and 6%. During the execution of the static load tests, the weather was cloudy with a temperature of 10 to 12 °C.

The static load test was carried out in two loading cycles, where the maximum stress value acting on a rigid circular load plate of 300 mm diameter was considered to be 0.20 MPa (the methodology of the static load test is characterized in more detail in, e.g., [4]). Geofiltex 63/20 T non-woven separating geotextile was applied to the surface of the subgrade surface to protect the overlying coarse-grained material from mixing with the fine-grained material of the subgrade surface or the coarse-grained sharp-edged material of the sub-ballast layer from pressing into the fine-grained (low-deformation-resistance) subgrade surface. Subsequently, a 150 mm layer of 0/63 mm foam glass was applied as a thermal insulation layer. On its surface, the static modulus of deformation $E_{def2}$ = 20.60 MPa and the compaction $E_{def2}/E_{def1}$ = 2.12 were determined (using the same methodology as for the subgrade surface).

A Sisgeo L141 pressure transducer (Figure 3), consisting of two stainless-steel plates welded around their circumference, was then mounted on the surface of the foam glass aggregate. The annular space between these plates was filled with vacuum-vented oil and then connected to the transducer via a stainless-steel pipe, thus forming a closed hydraulic system. The Sisgeo L141 pressure transducer allows measurements from 0–200 kPa to 0–10 MPa, with a sensitivity of 0.002% FS (full scales), where FS = 200 kPa. The measurement can be carried out in a temperature range from −20 °C to +80 °C.

An acceleration sensor, the IEPE A/140/SC single-axis accelerometer (Figure 4), made of stainless steel, was then mounted close to the pressure sensor, allowing long-term analysis of vibration propagation in an adverse environment. The nominal sensitivity of the sensor is 100 mV/g ± 10%. The typical frequency response is 1 to 10 kHz ± 5%. Measurements can be performed over a temperature range from −50 °C to +120 °C.

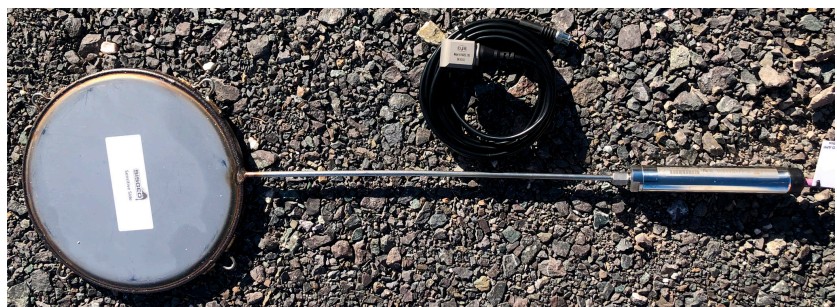

**Figure 3.** Pressure transducer Sisgeo L141 [24].

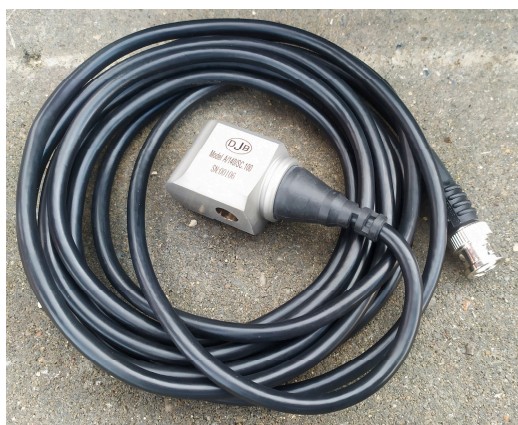

**Figure 4.** Single-axis accelerometer IEPE A/140/SC [25].

Figure 5 demonstrates how the tested sensors were mounted (the photo was taken after the measurements were completed as part of dismantling the experimental field). The mounting of the sensors (accelerometer on the left, pressure sensor on the right) was carried out on a thin buffer layer of sand to ensure complete surface contact with the structural layer of foam glass. The pressure sensor has four metal eyelets along the outside (see Figure 3), which allows for stabilizing its directional position on the structural layer by pushing the locking pins into the layer. The accelerometer has an opening for this purpose (see Figure 4) to positionally stabilize the sensor.

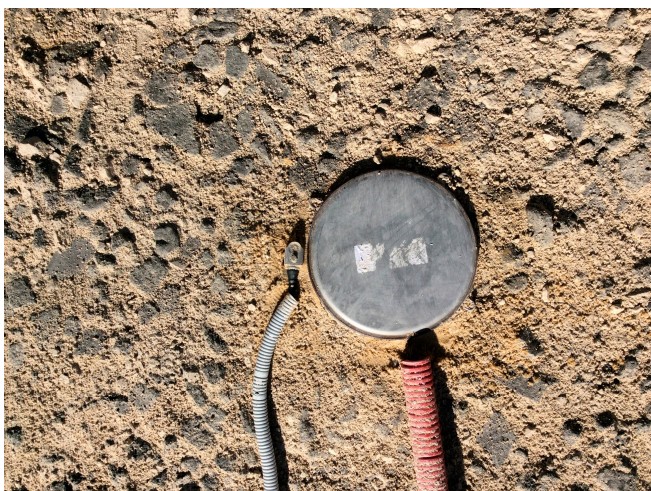

**Figure 5.** Mounting the sensors on the surface of foam glass aggregate.

After the sensors were installed, the experimental field was subsequently completed by applying a 220 mm layer of crushed aggregate, fr. 0/31.5 mm. The static modulus of

deformation at the surface of the crushed aggregate layer was $E_{def2}$ = 48.90 MPa (Figure 6), and the compaction $E_{def2}/E_{def1}$ = 1.45 (the maximum value allowed for coarse-grained materials is 2.60). The compaction of the crushed aggregate layer was implemented with a lightweight vibratory plate.

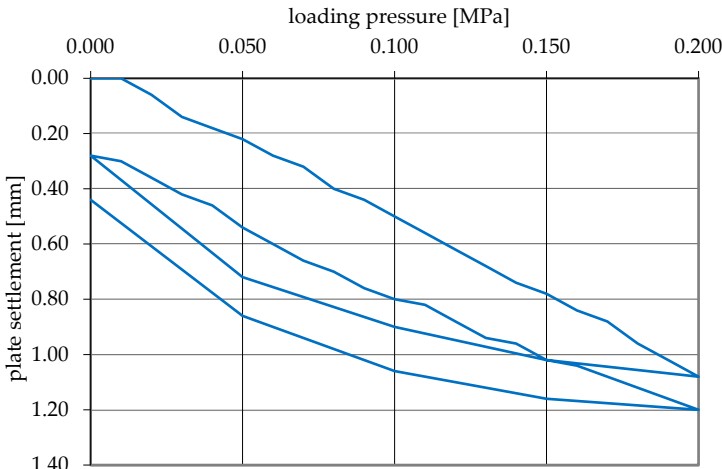

**Figure 6.** Output of the static test performed on the surface of the crushed aggregate layer of fr. 0/31.5 mm—plate settlement recorded at each change of 0.01 MPa in the loading pressure.

A series of dynamic load tests (12 measurements) were subsequently carried out on the surface of the crushed aggregate layer of 0/31.5 mm using the LDD 100 (Figure 7), for which the measurement methodology followed [26]. The LDD 100 (Lightweight Dynamic Plate Testing device) is a device that allows the deflection of a measured layer below the center of a rigid circular plate to be measured, caused by a damped shock pulse of approximately half-sinusoidal shape. This device automatically calculates the dynamic (shock) modulus (Figure 8) generated in the measured layer, determined as the average value of three measurements (application of 3 weight drops on a rigid circular plate).

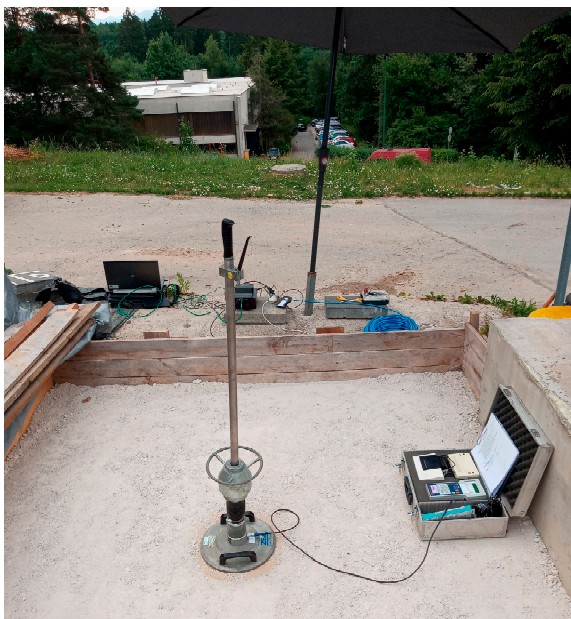

**Figure 7.** Implementation of dynamic load test using LDD 100 on the surface of crushed aggregate layer fr. 0/31.5 mm and monitoring of its dynamic response.

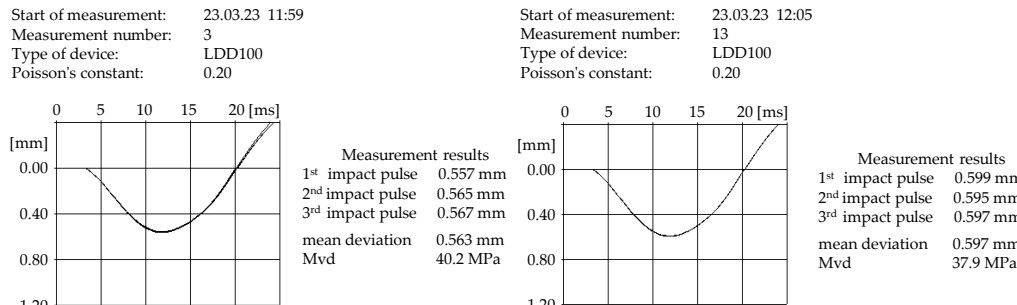

**Figure 8.** Immediate test results using LDD 100 on the surface of a 0/31.5 mm crushed aggregate layer and monitoring its dynamic response.

The first six measurements were performed with a recording frequency of 3200 Hz, and the others with a recording frequency of 6400 Hz. The dynamic load test applied a maximum impact force F = 7070 ± 70 N to a rigid circular load-bearing plate of diameter *d* = 300 mm. The impact pulse was induced by dropping a weight of mass 10 kg from a specified height onto a circular load plate equipped with a shock absorber, which distributed the force in a specified time interval. A contact stress σ = 0.10 MPa was generated under the circular load plate. It approximately corresponded to the railway loading (at the level of the sub-ballast upper surface). During the impact, stress was generated in the measured structural layer, which caused its (elastic or plastic) reshaping. The deflection caused was determined by a suitable track (load plate drop) sensor. The measurement methodology using the light dynamic load test and its possible applications is described in more detail in [27].

## 3. Results from Experimental Measurements and the Computational Model

The layout or position of the sensors used in the investigation of the dynamic response of the most significant structural layer, forming the basis for the construction of the track superstructure, was as follows:

- AC1—internal accelerometer in the LDD, located on the surface;
- AC2—a single-axis IEPE A/140/SC accelerometer located below the LDD at the interface, the interface between the LDD and the subgrade;
- TB2—a Sisgeo L141 pressure cell also located beneath the LDD at the interface between the monitored layer and the subgrade.

This sensor placement provides a relevant way to identify the properties of the modeled layer. The dynamic response was monitored at two locations. The first spot was at the surface, at the inlet directly below the axis of the LDD 100 loading device, and the second monitored location was between the above-mentioned layer of crushed aggregate and the subgrade. Considering the simplest material model, characterized by two basic mechanical quantities (*E*—dynamic elastic modulus; *ν*—Poisson's constant), it is sufficient to correlate the numerical experimental data at the AC2 outlet. Since the TB2 pressure cell was not placed under the load cell because its measuring range did not allow it, it is not possible to compare the input in terms of stresses. Thus, the comparison of the time records is irrelevant. For this reason, only the maximum peak voltages $\sigma_z$ can be compared in the conclusions.

From the point of view of the nature of the dynamic action, it is an impact load. These occurrences happen quickly, so the graphical depiction will display values in seconds (s) or milliseconds (ms) on the time axis. All measurements were implemented in the form of a series of single pulses, initiated using a weight of 10 kg. In the loading apparatus, the weight fell from a specified height (determined by the calibration of the apparatus) directly onto the circular surface of the LDD 100. A total of 12 series of measurements were performed during the entire experimental measurement campaign. Each of them

contained at least three input pulses from the loading device. An example of one series of measurements at the AC1 input is demonstrated in Figure 9.

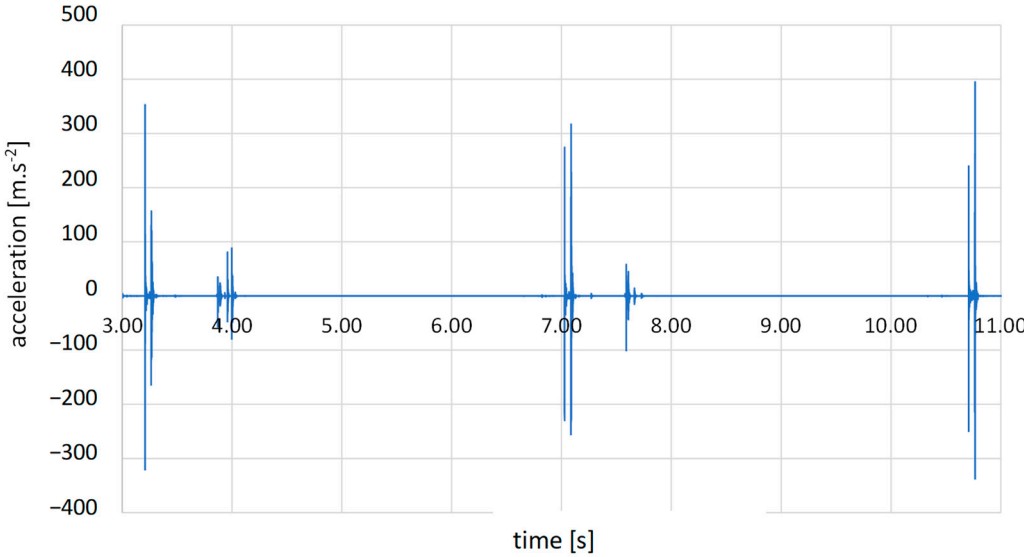

**Figure 9.** Sample of measurement No. 11—series of three pulses.

For the recording of the individual pulses, a single-axis sensor IEPE A/140/SC and a PULSE measurement system by Brüel & Kjær [28] were used. It is a sophisticated system designed to measure and evaluate vibrations propagating from different vibration sources. Due to the well-established wave process theory, the vertical component of the vibration is dominant, and a uniaxial sensor is thus sufficient. Subsequently, each of these 12 measurements was independently evaluated in terms of the achieved acceleration and deflection over time using Sigview version 3.0.2.0, a system designed for signal analysis.

### 3.1. Computational Model of the Experimental Stand

The numerical model was developed based on the finite element method (FEM) in the VisualFEA computing system version 2020 [29]. The software is used to analyze physical problems that may arise in various fields of science and technology. With its help, it is possible to evaluate different types of static analyses and dynamic phenomena. After finite element analysis data are evaluated, the results are visualized in various forms.

The whole numerical model has the shape of a cube (2.5 m × 2.5 m × 0.35 m) divided into two layers; their interaction was not included in the actual calculation. This shape was selected on the basis of the considerable attenuation properties of the materials under study. The attenuation characteristics were included in the calculation in the form of Rayleigh coefficients. Each modeled layer has its geometrical and mechanical properties set. In the first one, the crushed aggregate layer has a thickness of 220 mm, a dynamic modulus of elasticity (E) of 50 $MN \cdot m^{-2}$ and Poisson's constant $\nu = 0.30$. The second layer—the subgrade—simulates a layer of infinite thickness. In this case, a thickness of 150 mm was selected, due to the research already carried out in this area published in several publications, e.g., [30,31]. The mechanical properties are characterized by a dynamic modulus of elasticity (E) of 20 $MN \cdot m^{-2}$ and Poisson's constant $\nu = 0.10$. The individual elements of the finite element model (FEM) of the considered computational model have the shape of prisms with a triangular base. Each prism has six connection nodes, and the total number of model elements is 1400. The bottom wall of the model is elastically supported with stiffness in all directions $k_{xyz} = 1000$ $(MN/m)/m^2$ (Figure 10).

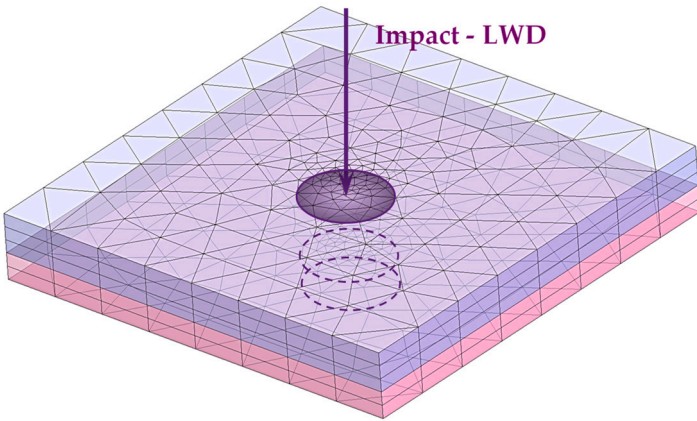

**Figure 10.** Numerical model of the experimental measurement.

In the upper part of the model, a circular layer was created in the middle of the plate running through both parts, simulating the LDD 100 device with a load plate diameter of $d = 300$ mm, whose function is to input shock pulses to the whole numerical model. Around this region, the finite element mesh is densified in such a way that the proportionality of the model is preserved, given the results of previous research analyzed in [32,33].

### 3.2. Dynamic Load of the Computational Model

Upon loading the initial model, we obtained the measured acceleration and deflection data corresponding to the single points on the LDD 100 device load plate area indicated in Figure 10. For each series, the pulse with the largest deflection and acceleration amplitude was selected. Measurements deemed non-relevant were excluded from the statistical set due to irregularities in the waveform caused by the environmental inhomogeneity.

The remaining seven measurements and their dynamic loads are summarized in the form of plots in Figure 11. Only the most relevant part of the record at the entry point AC1 is shown, due to further evaluation and analysis. The vibration finishing of the whole system has not been considered in terms of loads. The left part of Figure 11 shows the primary vibration acceleration records obtained in the experimental measurements. The right-hand side contains graphs representing the displacement of the measured point AC1. This relocation was achieved by numerical integration of the primary records of the acceleration of vibration at point AC1.

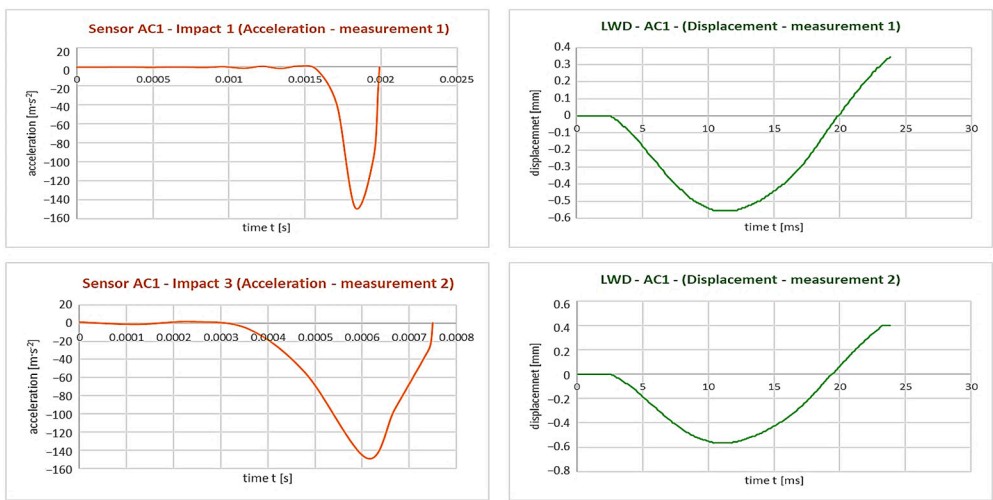

**Figure 11.** *Cont*.

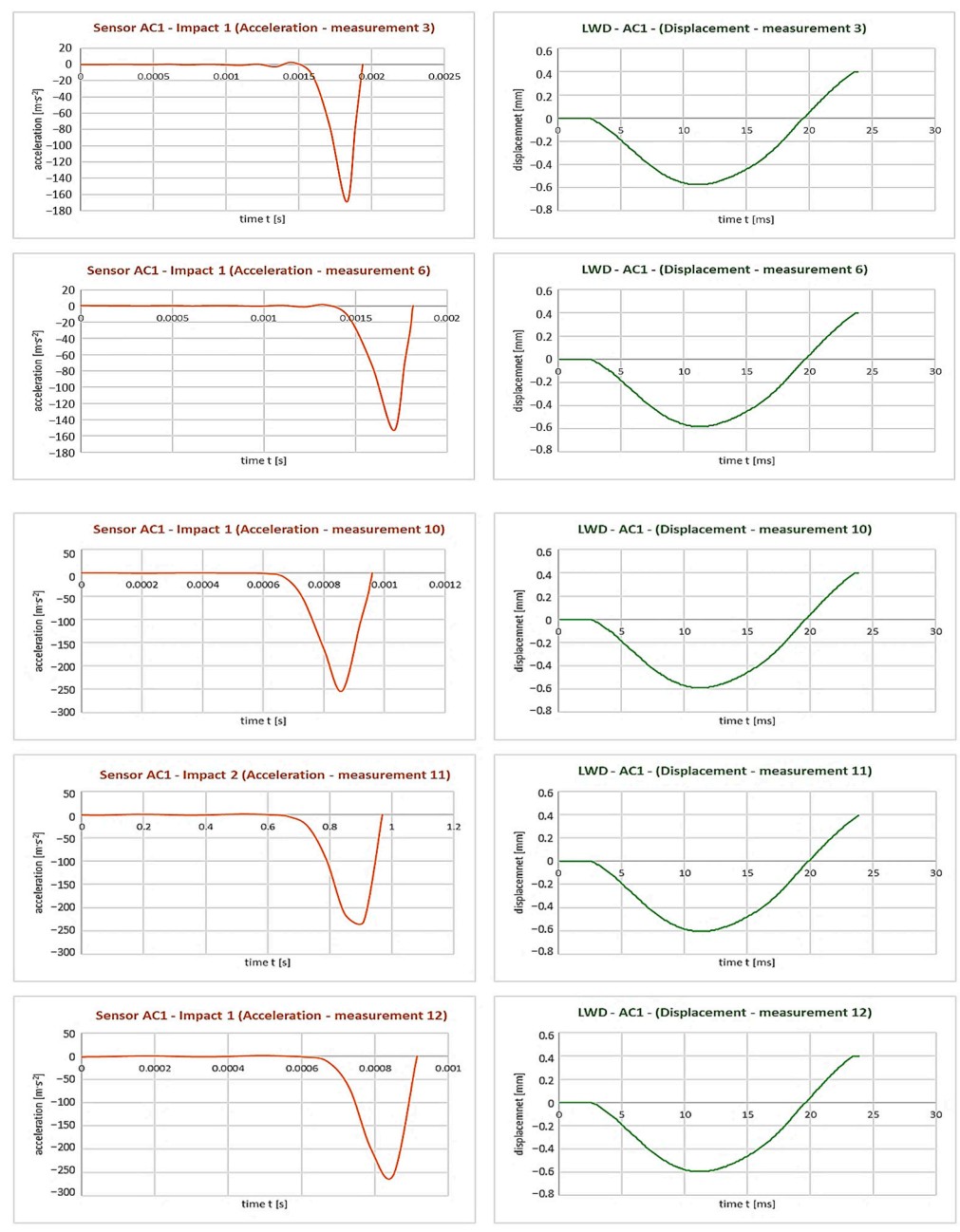

**Figure 11.** Measured accelerations and deflections over time for AC1-LWD—measurements 1, 2, 3, 6, 10, 11 and 12.

### 3.3. Dynamic Response at the Interface of Computational Model Layers

The analysis aimed to obtain the response of the computational model to the experimentally defined load in AC2—the layer between the crushed aggregate and the subgrade. By sensitivity analysis and changing the basic mechanical parameters, algorithmization and stepping of the calculations, the system with optimal mechanical properties was identified. This iterative procedure was concluded at the highest agreement of experimental and numerical response at point AC2. An example of the result of this algorithm is presented in Figure 12.

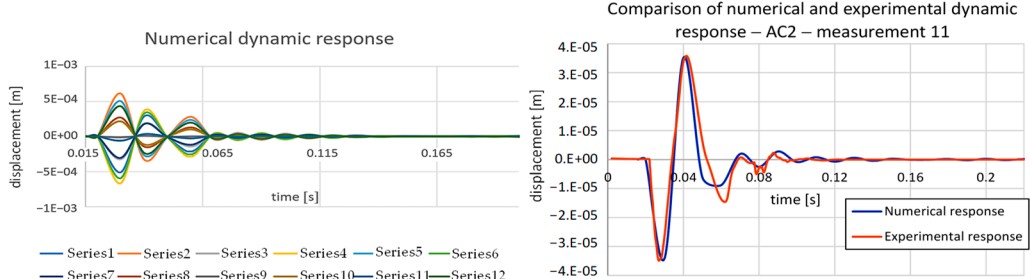

**Figure 12.** Numerical dynamic response (**left**) and its comparison between numerical and experimental results—measurement 11 (**right**).

Figure 12 demonstrates the resulting numerical response of the FEM model to dynamic loads applied to the circular surface of the modeled experimental stand. Since this is a numerical analysis, the response was evaluated in a point field around the center of the circular surface. This point field was near the position of the AC2 sensor at the experiment and consisted of 12 nodes of the FEM network. Therefore, the top of Figure 11 is the pooled graph for the 12 vibration deflection waveforms. The tuning optimization algorithm evaluated measurement 11 as the most relevant pulse series. It is the measurement in which the deflection amplitudes reach their maximum values at the input-sensor AC1. The orange color in the right part of Figure 12 shows the experimentally obtained response, and the blue demonstrates the numerical response. The experimental response is obtained by measurement, and the numerical response is the averaged value of the records from the left graph of Figure 12. The left part of the Figure 12 shows the individual numerical dynamic responses of the system. The amplitude analysis of the maximum values for the load data input to the algorithmized calculation is provided in Table 1.

**Table 1.** Evaluation of maximum deflection amplitudes for all measurements and measured points under investigation.

| Measurement Series | Sampling Frequency | LWD—Displacement—Sensor AC1 | | | Layer—Displacement—Sensor AC2 | | | Acceleration—AC2 |
|---|---|---|---|---|---|---|---|---|
| | | 1st Impact | 2nd Impact | 3rd Impact | 1st Impact | 2nd Impact | 3rd Impact | 1st Impact |
| No. | [sam/s] | $v(t)_{max}$ [mm] | | | $\overline{v(t)_{max}}$[mm] | | | $\alpha(t)_{max}$ [mm·s$^{-2}$] |
| 1 | 8192 | 0.554 | 0.554 | 0.554 | 0.083 | 0.0817 | 0.0826 | 148.281 |
| 2 | 8192 | 0.554 | 0.562 | 0.562 | 0.0852 | 0.0862 | 0.0855 | 149.045 |
| 3 | 8192 | 0.569 | 0.577 | 0.577 | 0.0858 | 0.0854 | 0.0873 | 169.151 |
| 4 | 8192 | 0.585 | 0.569 | 0.577 | *** | *** | *** | 151.181 |
| 5 | 8192 | 0.569 | 0.569 | 0.577 | *** | 0.0839 | *** | 148.76 |
| 6 | 8192 | 0.577 | 0.592 | 0.592 | 0.0856 | 0.0935 | 0.0875 | 153.042 |
| 7 | 16,384 | 0.577 | 0.569 | 0.577 | 0.0847 | 0.0738 | 0.0853 | 237.025 |
| 8 | 16,384 | 0.585 | 0.585 | 0.577 | 0.0865 | 0.0842 | 0.0817 | 202.413 |
| 9 | 16,384 | 0.577 | 0.592 | 0.585 | 0.0777 | 0.0877 | 0.0884 | 214.969 |
| 10 | 16,384 | 0.585 | 0.585 | 0.592 | 0.0854 | 0.0846 | 0.0864 | 254.013 |
| 11 | 16,384 | **0.6** | **0.592** | **0.6** | 0.0883 | 0.0869 | 0.0871 | 229.745 |
| 12 | 16,384 | 0.592 | 0.592 | 0.592 | 0.089 | 0.088 | 0.0891 | 255.781 |
| Σ | - | 6.924 | 6.938 | 6.962 | 0.8512 | 0.9359 | 0.8609 | *** high noise |
| Average | - | 0.577 | 0.578 | 0.580 | 0.0851 | 0.0851 | 0.0861 | |

In the resulting FEM model, the parameters of the mechanical properties of the two layers were optimized. The elastic moduli *E* and Poisson's coefficients *v* for the simplest material type to model—the linear elastic isotropic tuning optimization module—were set to the values specified in Section 3.1. The bulk densities of the individual layers (easily measurable on the specimens) and the stiffness of the support of the bottom layer defined from the static test of the subgrade by the correlation relations given in [5] were stable.

As an illustration of the regular deformation behavior of the whole computational model, Figure 13 demonstrates the deformation of the maximum impulse at the second time step, i.e., at the second millisecond of the vibration time history of the system. The

other deflections decrease exponentially due to the attenuation, and since the vibration is harmonic in nature, it is not meaningful to show the deflections in the other time steps.

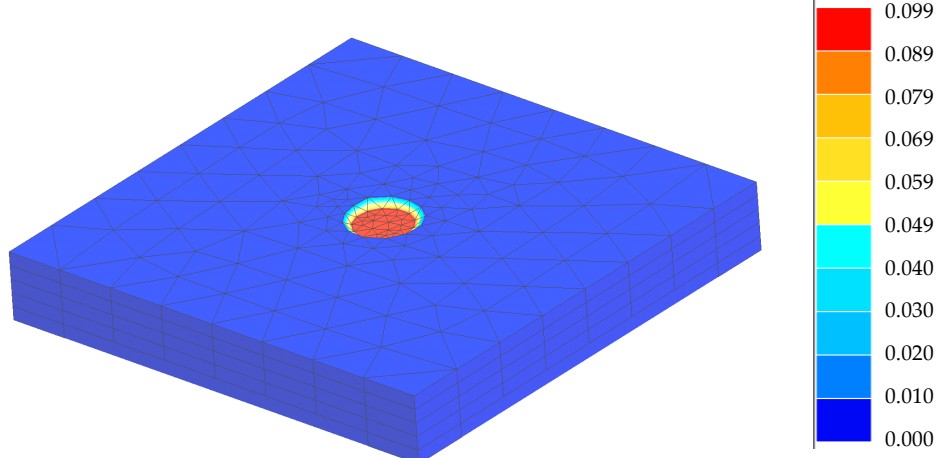

**Figure 13.** Deformation of the overall computational model for measurement 11, 1 pulse, time step 2.

## 4. Conclusions

Act No. 17/1992 Coll. on the environment [34] defines sustainable development as development that maintains the ability of present and future generations to meet their fundamental life needs without compromising the diversity of nature and the essential functions of ecosystems. Notably, among various transport systems, rail transport has the least detrimental impact on environmental quality, thus contributing to the preservation of natural ecosystems. Nevertheless, there is ample room for rail transport to further enhance its environmental friendliness and minimize adverse effects on sustainability. Also in connection with the gradual depletion of natural building materials used in the sub-ballast layers of the railway track, the necessary optimization of its structural composition and the design of the track skeleton, it is necessary to quantify and qualify the actual effects of rolling stock on the roadway thus designed. This can only be achieved by high-quality sensors, recording technology and relevant evaluation of the measurement values obtained.

An optimized calculation model, tuned based on relevant measurements, can improve the design of the individual layers in the railway track structure. Algorithmization of numerical calculations, in parallel with the correlation of measurement results and numerical results, will allow, based on simple measurements and even with a smaller number of sensors, the optimization of the mechanical properties of the individual structural layers. The paper demonstrates this approach by tuning the two basic mechanical parameters of the simplest material model for two structural layers. It will allow the application of various new, possibly simultaneously environmentally friendly, materials to the most exposed structural components of the sub-ballast layers.

The obtained results of the experimental measurements on the DRETM experimental stand and the numerical modeling indicate the following specific conclusions:

1.  The input data obtained experimentally in the form of accelerograms and integrated LWD vibration deflection records are consistent regarding the sufficient number of measurements, and they demonstrate a high degree of correlation for the two fundamental values of sampling frequencies—both 8192 Hz and 16,384 Hz. This fact is characterized in Section 3.2.
2.  The suitability of using simple material models and finite element models of geometrically simple shapes is also demonstrated due to the high attenuation of the optimized structural sub-ballast layers. It is also appropriate to model simplified kinematic boundary conditions and to do so in conjunction with the results of static subgrade load tests (Sections 2 and 3.1).

3. The most significant study finding is the high applicability of algorithmization of the tuning approach to tracking and adjusting multiple mechanical properties of the modeled layers. This result is declared in Section 3.3. Although the idealized system does not perfectly correlate with the one measured in the actual stand, the tuned model can be considered relevant, and its results are applicable for long-term monitoring and diagnosis of the stiffness parameters of the most significant sub-ballast layers.

4. A non-negligible research contribution is the verification of the possibility of using a limited number of sensors while the experimental results are nevertheless applicable.

The results obtained from the DRETM experimental stand and its FEM model can be considered useful for practice. Monitoring cells with embedded sensors are currently being built on actual track sections in the way presented in this study. Their aim is to monitor changes in the basic mechanical properties of the single layers and define the condition when these layers need to be replaced. This is of great importance for the ecology of the railway infrastructure environment, as structural layers are often applied according to (outdated) regulations and not based on the latest measurement and research results. Modern approaches should be incorporated into legislative documents today, as they can efficiently, economically and, above all, in an environmentally friendly way optimize the reconstruction or modernization work on the railway infrastructure of any railway manager. In the near future, it is planned to apply sensors to structural elements of the railway superstructure in the test sections to monitor the influence of various superstructure modifications (application of sub-sleeper pads, reinforcing rails, bonding of the ballast bed). Obtaining actual input parameters will also be the basis for numerical modeling and design of modified structural compositions of the transition zones between the artificial structures and the earthwork.

**Author Contributions:** Conceptualization, L.I., P.D. and M.M.; methodology, P.D. and M.M.; software, Z.P.; validation, L.I., P.D. and Z.P.; formal analysis, P.D., M.M. and Z.P.; investigation, P.D. and M.M; resources, L.I.; data curation, P.D. and Z.P.; writing—original draft preparation, L.I., P.D. and Z.P.; writing—review and editing, L.I., P.D. and Z.P.; visualization, P.D. and Z.P.; supervision, L.I.; project administration, L.I.; funding acquisition, L.I. All authors have read and agreed to the published version of the manuscript.

**Funding:** This work was supported by the VEGA grant project 1/0084/20 Numerical and experimental analysis of transition areas of objects of structures of railway superstructures and objects of formation substructure.

**Institutional Review Board Statement:** Not applicable.

**Informed Consent Statement:** Not applicable.

**Data Availability Statement:** Data are contained within the article.

**Conflicts of Interest:** The authors declare no conflicts of interest.

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
