# Peer review of "Experimental Monitoring of Dynamic Parameters of the Sub-Ballast Layers as a Prerequisite for a High-Quality and Sustainable Railway Line"

_sustainability, doi:10.3390/su16062229_

Round 1

Reviewer 1 Report

Comments and Suggestions for Authors

The paper adopts on-site experiments and finite element methods to study  dynamic parameters of the Sub-Ballast layers, which has certain significance for evaluating the environmental impact of railways.

Comments:

(1) The actual situation of on-site experiments in the paper needs to be further clarified, especially lacking some on-site photos. For example, it is possible to consider adding actual photos of the site in Figure 2.

(2) How are the boundary conditions of the model determined? Especially when the impact load is applied in a circular shape with a square boundary. In addition, thickness has a significant impact on load transfer, why did the model only take 0.35m?

(3) The conclusion of the paper needs to be refined.

Comments on the Quality of English Language

The paper adopts on-site experiments and finite element methods to study  dynamic parameters of the Sub-Ballast layers, which has certain significance for evaluating the environmental impact of railways.

Comments:

(1) The actual situation of on-site experiments in the paper needs to be further clarified, especially lacking some on-site photos. For example, it is possible to consider adding actual photos of the site in Figure 2.

(2) How are the boundary conditions of the model determined? Especially when the impact load is applied in a circular shape with a square boundary. In addition, thickness has a significant impact on load transfer, why did the model only take 0.35m?

(3) The conclusion of the paper needs to be refined.

Author Response

Thank you for reviewing our article. We attach a document with responses to the reviewer's comments.

Reviewer 2 Report

Comments and Suggestions for Authors

The Chapter 1 (Introduction) is relatively long. It could be split into two chapters (short introduction and then the review of existing works)

Short descrition of LDD 100 light dynamic plate would be useful for clarity (222).

Please explain what „the environmental inhomogeneity” precisely mean ? (313-314)

Short comment on AC1 displacement (Fig. 11, lines 319-324) is necessary.

The quality of Figure 11 could be improved. Moreover it would be advisable to present time (horizontal axis) in the same units for acceleration and for deflection (now [s] and [ms] respectively)  

Figure 12 requires correction of description – use „series” instead of „rady” (334-335).

Are sampling frequencies (8192 Hz and 16384 Hz) related to the settings (or characteristics) of measuring equipment ?

What about verification of the models for different values of thickness of particular layers?

Comments on the Quality of English Language

roadway or permanent way?

Author Response

Thank you for reviewing our article. We are enclosing our responses to the reviewer's comments.

Reviewer 3 Report

Comments and Suggestions for Authors

The article discusses the issues of sensors calibration and numerical model setup in the context of monitoring dynamic load transfer from train traffic to sub-ballast layers. I believe the paper is publishable, but it does require a revision and several clarifications that I consider essential:

1. There is no basic information about weather conditions during measurements e.g. temperature, soil moisture.

2. A comment is needed regarding the possible impact of weather conditions on the measurements. Will the same measurements taken in winter/summer/after rain give the same results, considering properties of the surface layers?

3. Rows 205-209. Description of the sensors mounting is not clear enough. Are they fixed to the foam glass aggregate mechanically or just placed on the surface?

4. Figure 11, right hand column of plots showing (vertical?) deflections over time. The displacement value [mm] decreases and then increases so that it is greater than in the initial moment of time t=0. Does this mean that the sensor changes its position before and after the measurement?

5. Figure 12 (right) shows comparsion between numerical and experimental response but there is no clear indication which one is the experimental result. The results are descibed as Rady1 and Rady2 which is the same label as in the left plot describing numerical results only. And Rady2 (orange) result in the right hand side plot has smaller amplitude that the same Rady2 result in the left hand side plot.

6. Why were only experimental and computational displacements compared but not the values of pressure and acceleration? The paper is too cursory in its description of the results verification.

7. On what basis were the sampling frequencies of 8192 Hz and 16384 Hz adopted?

8. Some dynamic links to cited articles lead to different articles e.g. citation [13] https://doi.org/10.1016/j.conbuildmat.2016.02.084. After clicking the address it opens as https://doi.org/10.3390/sym14030536.

Author Response

(The authors gave the same response as above.)

Reviewer 4 Report

Comments and Suggestions for Authors

In the manuscript titled “Experimental Monitoring of Dynamic Parameters of the Sub-Ballast Layers as a Prerequisite for a High Quality and Sustainable Railway Line”, the authors Libor Ižvolt et al. calibrate the sensors installed in the different test sections of the trans-European corridor number V. The result of the calibration was the demonstration of sufficient sensitivity of the sensors and their suitability for implementation in an actual railway track structure. However, there are still some problems with the content and formatting of the article, and the following are some of my specific suggestions for changes in the article, this manuscript may be accepted after minor modification.

Minor comments

1.     The images throughout the article are not clear, the authors should replace them with high-resolution images.

2.     In Chapter 3, the authors mention that “Monitoring cells with embedded sensors are currently being built on actual track sections in the way presented in this study.” The authors should add the future scope of the paper.

3.     In Chapter 3, the authors should add a schematic of the installation locations of three sensors.

4.     Defects in railroads can also cause damage to wheels, so authors should consider citing the following relevant studies to flesh out the introduction: 10.1016/j.measurement.2022.111268

5.     The authors should add a descriptive analysis of Figure 11, and Figure 12.

Comments on the Quality of English Language

No

Author Response

(The authors gave the same response as above.)

Round 2

Reviewer 1 Report

Comments and Suggestions for Authors

1. The title of the paper should be concise.

2. The abstract should show the main research results.

Comments on the Quality of English Language

The English quality of the paper is acceptable.

Author Response

  1. The title of the paper should be concise.

Thank you very much for your suggestion, but the title has been accepted by the editorial board of the special issue. It can no longer be changed at this phase.

  1. The abstract should show the main research results.

The abstract has been supplemented. Thank you for your comment.

Reviewer 3 Report

Comments and Suggestions for Authors

I have no more comments, I recommend publication.

Author Response

Thank you very much for the review.